# High-Performance Forecasting of Spring Flood in Mountain River Basins with Complex Landscape Structure

**Yuri B. Kirsta \* and Irina A. Troshkova**

Institute for Water and Environmental Problems SB RAS, Molodezhnaya 1, 656038 Barnaul, Russia
\* Correspondence: kirsta@iwep.ru

**Abstract:** We propose the methodology of building the process-driven models for medium-term forecasting of spring floods (including catastrophic ones) in the mountainous areas, the hydrological analysis of which is usually much more complicated in contrast to plains. Our methodology is based on system analytical modeling of complex hydrological processes in 34 river basins of the Altai-Sayan mountain country. Consideration of 13 types of landscapes as autonomous hydrological subsystems influencing rivers' runoff (1951–2020) allowed us to develop the universal predictive model for the most dangerous April monthly runoff (with ice motion), which is applicable to any river basin. The input factors of the model are the average monthly air temperature and monthly precipitation for the current autumn–winter period, as well as the data on the basin landscape structure and relief calculated by GIS tools. The established universal dependences of hydrological runoffs on meteorological factors are quite complex and formed under influence of solar radiation and physical–hydrological patterns of melting snow cover, moistening, freezing, and thawing of soils. The model shows the greatest sensitivity of April floods to the landscape composition of river basins (49% of common flood variance), then to autumn precipitation (9%), winter precipitation (3%), and finally, to winter air temperature (0.7%). When it is applied to individual river basins, the forecast quality is very good, with the Nesh–Sutcliffe coefficient NSE = 0.77. In terms of the accuracy of process-driven predictive hydrological models for the mountainous areas, the designed model demonstrates high-class performance.

**Keywords:** mountain rivers; runoff forecast; air temperature; precipitation; system-analytical modeling; Altai-Sayan



## 1. Introduction

Floods are dangerous natural phenomena, which seriously affect human economic activity worldwide [1]. For making timely reliable forecasts of floods on rivers under atmospheric processes unbalanced by climate change, various forecasting technologies and mathematical models with provision for relevant environmental factors are developed [2–4]. Present-day models for runoff forecasting can be divided into three groups (conceptual, physical, and empirical ones) [5] or two main types (process- and data-driven models) [6,7]. The first type involves various physical and hydrological models directly describing the processes of river flow formation by means of differential, algebraic, and other equations with related parameters. Being in fact the scientific method of hydrological research [4,8], these models adequately reflect dynamics of processes influenced by environmental factors. Models of the second type are based on mathematical processing of initial data and intended for practical use of river flow forecasts in water resources management and economic activities. These include autoregressive integrated moving average, multiple linear regression, artificial neural network, adaptive neuro-fuzzy inference system, extreme learning machine, generalized regression neural network, random forest, support vector machine, etc. [7,9]. Development of such models is not laborious; often they provide greater accuracy in forecasting river flows in comparison with physical-hydrological models. In

addition, classical statistical models characterizing the occurrence probability of floods of different intensity for local areas are still used [10,11].

Due to the growing popularity of data-driven models, it is reasonable to consider, for example, a successful model designed by Q. Wang and co-authors for monthly streamflow prediction [7]. It is based on the embedded feature selection method with the improved gray wolf optimizer and support vector machine ensuring the achievement of a high prediction accuracy (Nash–Sutcliffe efficiency of 0.82) on average for all months of the year. The available data on monthly flows for relevant previous months (best correlated with predicted runoff) were used.

At the same time, it is obvious that winter antecedent flows cannot correlate with spring ones because of sharp differences between the hydrological regime of frozen mountain rivers and that of accumulated snowmelt and ice drift in early spring (April). Therefore, data-driven models [5,7] fail to achieve the maximum possible accuracy in forecasting spring floods on mountain rivers.

Top-quality predictive models require an adequate description of hydrological processes throughout the catchment area that is hardly feasible without the detailed spatially distributed data on precipitation, air temperature, vegetation and landscapes, underground aquifers, relief, soil properties, river channel profile, etc. For the mountain areas, such information is mostly absent that greatly hampers models development and brings to poor accuracy both of river flow models [12] and flood forecasts [13]. However, forecasts for mountainous areas are extremely important because of the increasing threat of catastrophic floods [14] able to induce overflow of hydroelectric reservoirs or disrupt water supply to local population [15–17]. Despite the growing number of studies on this topic, there are still no universal methods and models for high-quality medium-term or long-term forecasts of spring floods in the mountains.

In this paper, we solve the problem of creating a high-performance process-driven model for a medium-term forecast of spring floods in the mountainous areas. Using optimization methods for solving equations, we analyze a large sample of April monthly river runoffs (with ice motion) influenced by autumn–winter temperatures along with precipitation, and thus pursue an adequate reflection of hydrological processes in the mountain geosystems (landscapes) considered as autonomous hydrological subsystems of the catchment areas. We understand adequacy as a conformity of the model to physical principles and laws complemented by appropriate assumptions [18]. Our long-term investigations resulted in the development of the effective methodology for modeling complex natural systems with insufficiently studied processes and their unclear dependence on environmental factors [19,20]. This methodology is called system analytical modeling (SAM). The structure and values of parameters of the sought-for model are defined via solving the inverse mathematical problem for the large-dimensional system of equations; the data on the long-term parallel observations of the dynamics of the study characteristics and changing environmental factors are employed. Thus, SAM enables us to take into account the information contained in parallel observations implicitly. In our case, we analyze the first most dangerous month (April) of spring–summer floods of 34 mountain rivers of the Altai-Sayan mountain country (hereinafter referred as spring flood). Using a large number of river basins in solving the inverse problem simultaneously, we cancel out the influence of their individual hydrological features. Therefore, the created model describes common dependences of spring flood (with ice motion) on the environmental factors that make it universal.

## 2. Materials and Methods

### 2.1. Study Area and Database

For research, we selected 34 river basins located on the territory of the Altai-Sayan mountain country within 50° and 56° N, 83° and 100° E (Figure 1). The country is a part of the world watershed between the humid region of the Arctic Ocean and the arid drainless one of Central Asia. Its climate is sharply continental with long winters and short summers.

In XI–III months of the year, average monthly air temperatures are below 0 °C. Mountain ranges reach 2000–4500 m a.s.l. The largest river runoff is confined to the warm period of the year. In spring, runoff regime depends mainly on melting snow accumulated during winter [21,22]. In the Altai-Sayan mountain country, snow cover thickness can reach 2–3 m at altitudes of 1600–1700 m a.s.l., which brings destructive spring floods [23] (p. 120).

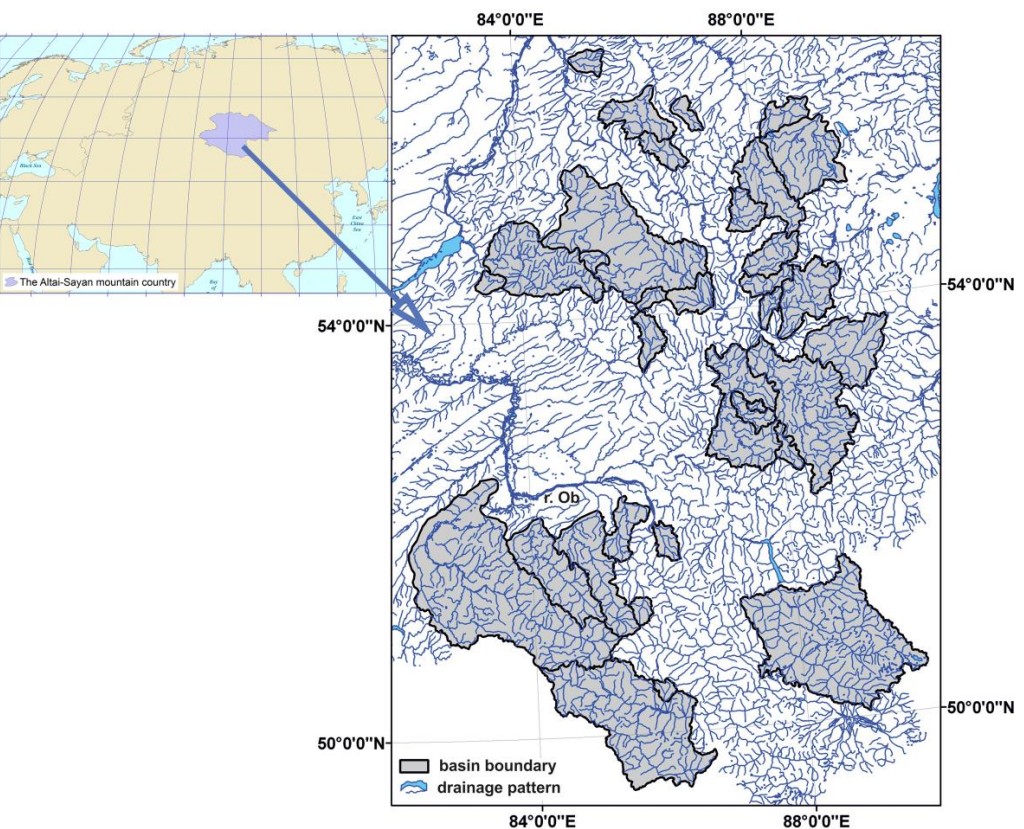

**Figure 1.** Map scheme of the location of 34 model river basins of the Altai-Sayan mountain country [24].

The Altai-Sayan mountain country is distinguished by a wide variety of geosystems and climatic conditions. Table 1 demonstrates the geosystems of the country divided into 13 typological groups (landscapes) by altitudinal belt and structural-tier heterogeneity of the territory, which affects their hydrological processes. Each of the 34 river basins were split into drainage zones/landscapes with own typical hydrological regime. Water runoff from these zones was accumulated at the river basin outlets, where the resulting river runoff was measured. The selected basins differed significantly both in landscapes and in areas (177–21,000 km$^2$).

To build a forecast model for spring runoff, we used the data of the Russian Federal Service for Hydrometeorology and Environmental Monitoring (Moscow, Russia) on river runoff (with ice motion) for April 1951–2020. The average runoff for all basin outlets was about 100 m$^3$/s. We supplemented runoff observations with the values of monthly precipitation and average monthly air temperatures, the GIS information (Institute for Water and Environmental Problems SB RAS, Barnaul, Russia) on landscape structure of the territory, the area, and altitude of landscapes in each river basin.

To perform SAM, homogeneous large samples of the data on analyzed characteristic dynamics and affecting environmental factors are needed. Therefore, the observed stream flows of 34 rivers were normalized to their long-term average for each basin. By transition from measuring flows in m$^3$/s to dimensionless units, we created a single homogeneous sample of runoffs for all rivers over the 70-year period 1951–2020. Landscape areas (km$^2$) in each basin were also converted to shares/percentages via dividing by each basin area.

**Table 1.** Identified types of landscapes of the Altai-Sayan mountain country [25].

| Landscapes (Geosystem Groups) |
| --- |
| 1. Glacial–nival high mountains with permafrost |
| 2. Goletz alpine-type high and middle mountains, pseudogoletz low mountains with permafrost |
| 3. Tundra–steppe and cryophyte–steppe high mountains with permafrost |
| 4. Forest high middle and low mountains |
| 5. Exposure foreststeppe and steppe high and middle mountains |
| 6. Forest–steppe, steppe low mountains and foothills |
| 7. Intermountain depressions with different steppes and forest–steppes |
| 8. Steppe and forest–steppe piedmont |
| 9. Nondrainable and intrazonal landscapes with partial permafrost |
| 10. Mountain river valleys |
| 11. Lowland river valleys |
| 12. Forest high and piedmont plains |
| 13. Aquatic landscapes |

Following the processing of river stream flows, we normalized the monthly values of meteorological factors. Since there were no regular meteorological observations in most of the selected basins, we calculated long-term monthly dynamics of normalized temperatures and precipitation uniform for the Altai-Sayan mountain country. Dynamics were identified from the data of 11 reference weather stations using the method of normalization and spatial generalization of meteorological characteristics [26,27]. These stations were located outside the basins, but had a continuous series of certified observations. We expressed dynamics in fractions/percentages of three corresponding long-term average monthly values 'in situ', i.e., the long-term average January temperature for months X–IV, the long-term average July temperature for months V–IX, and the long-term average July precipitation for all months of the year. Dynamics did not depend on the coordinates or altitude of site location, and was the same for all 34 river basins. Thus, the dependence of monthly temperature and precipitation on orographic factors, which form a contrasting climate picture in the mountains [28], was removed. It should be noted that in different parts of the mountain river basin, rain, snow, or zero precipitation can occur simultaneously and snow cover may melt or accumulate (Figure 2). This situation is typical for the analyzed territory: the northern slopes at altitudes over 3000 m receive 1200–2500 mm of precipitation per year, the middle parts of the slopes receive up to 600 mm, and the bottom receives about 200 mm [29].

In general, the database for SAM implementation included the following characteristics: hydrological (1747 normalized April runoff values for individual years 1951–2020), meteorological (840 + 840 = 1680 normalized monthly values of air temperature and precipitation), and landscape (352 area and altitude values calculated using ArcGIS 10.2).

### 2.2. Methodology of System-Analytical Modeling

We apply the SAM (Figure 3) methodology to identify and quantitatively characterize the functional relationships of April monthly floods with meteorological factors, morphometry, and the landscape structure of river basins. SAM is based on the analysis of natural systems' behavior as an integral complex of processes influenced by environmental factors [19,20,24,30]. This complex is described as a system of algebraic, differential, or other equations (i.e., the actual model), and the inverse mathematical problem is solved to determine the values of equation parameters. Solving this problem with a large array of experimental data ensures simultaneous quantification of interactions of the described processes and factors. To this end, we use optimization methods, which are relevant to large-dimensional systems of equations and provide minimum standard deviation (residual) of the calculated dynamic characteristic from the observed ones.

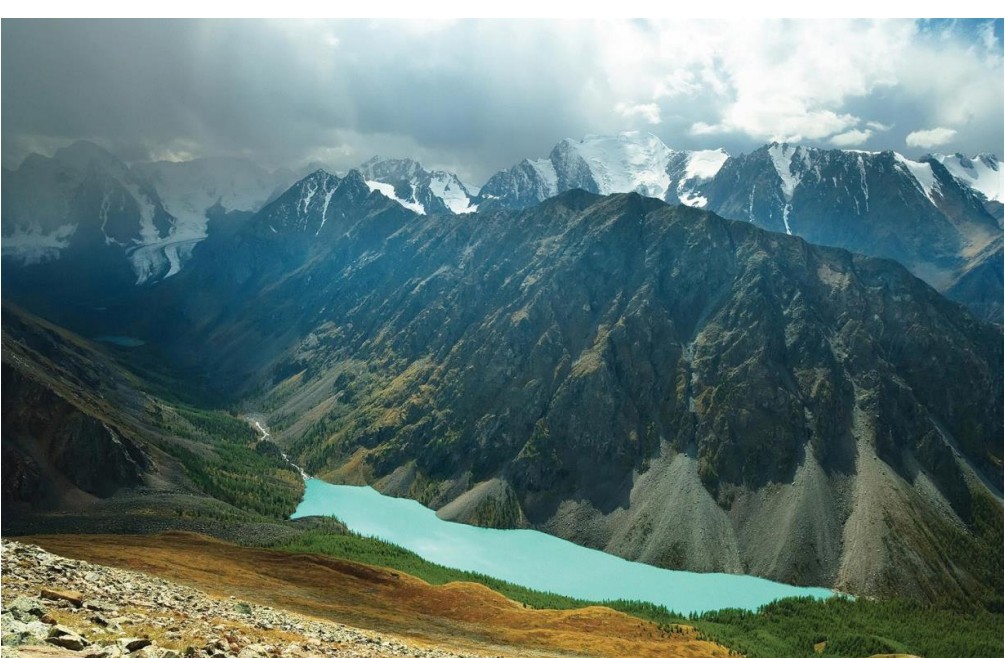

**Figure 2.** In the mountains, snow, rain, and zero precipitation may occur concurrently at different altitudes (the photo by unknown author shows the Shavla River basin in the Altai Mountains).

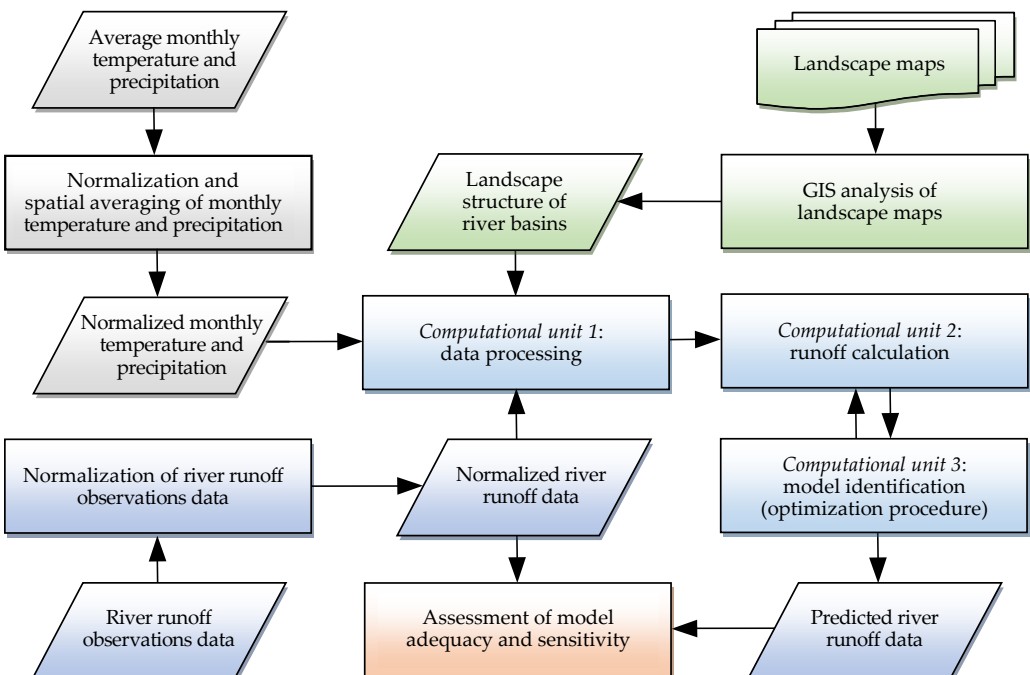

**Figure 3.** Flow chart of SAM methodology for building the process-driven hydrological models.

In the course of SAM, different versions of equations (which satisfy physical, chemical, biological, and other known laws) for describing the studied natural system are tested. The final version with the least discrepancy and sought equation parameters is considered the most adequate model of the system. It is important that SAM implementation does not require field measurements of many specific hydrological or soil characteristics of the study area. Just as in regression analysis, SAM needs a 5–10-fold excess of observational data over the number of model parameters. As a result, we can characterize processes and determine parameter values even more accurately than at their experimental measurement.

SAM is carried out in the well-known MATLAB programming environment, which provides processing of large data arrays (tens of thousands of values) and optimization solution of the inverse mathematical problem for the large-dimensional systems of equations with simultaneous calculations of up to 100 model parameters. When describing the insufficiently studied effect of environmental factors on the simulated processes, we use a continuous function $H$ consisting of three linear fragments:

$$H(X1, X2, Y1, Y2, Z1, Z2, X) = \begin{cases} Y1 + Z1(X - X1), & \text{if} & X < X1 \\ \frac{Y2 - Y1}{X2 - X1}(X - X1) + Y1, & \text{if} & \begin{cases} X1 \leq X < X2 \\ X1 \neq X2 \end{cases} \\ Y2 + Z2(X - X2), & \text{if} & X \geq X2 \end{cases}, \tag{1}$$

where $X1$, $X2$, $Y1$, $Y2$, $Z1$, and $Z2$ are parameters of sought dependence $H$ on input factor $X$ (Figure 4).

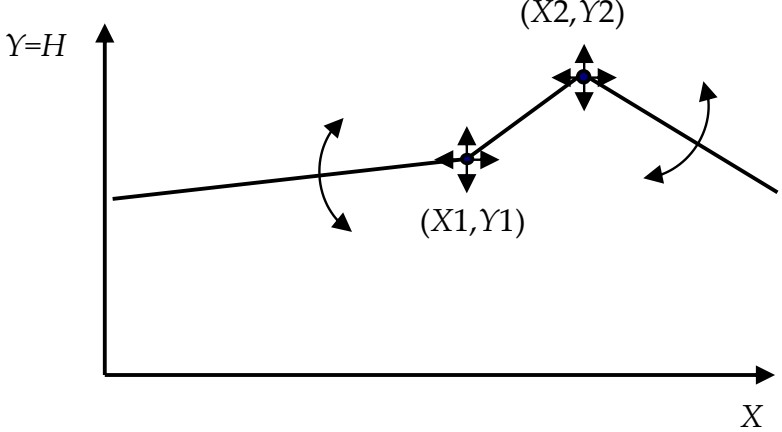

**Figure 4.** Continuous function $H(X1,X2,Y1,Y2,Z1,Z2,X)$ consisting of three linear fragments with arbitrarily changing parameters in Equation (1) [24].

Function $H$ describes almost any dependence between model variables for different parameter values. Using $H$, we do not impose artificial restrictions either on a form of the dependencies being checked nor parameter values. After solving the inverse problem and identifying the parameters, $H$ adequately reflects (simulates) the actual dependences of natural processes on environmental factors. Hence, SAM is the system simulation methodology.

The foregoing constitutes the first stage of SAM, which is commonly referred to as model identification. Being the most difficult stage in modeling the complex natural systems because of their insufficiently studied processes, it is marked by simultaneous identification of the structure, functional dependencies, and parameters of model. All of this applies to hydrology of the Altai-Sayan mountain country, with its complex terrain orographic structure, diverse hydrological processes, and contrasting climatic characteristics.

At the second stage of SAM, the developed model goes through verification via using the independent data that, in fact, is an assessment of the model universality. In SAM, the stage of verification is formal since a significant excess of experimental data over parameters ensures the universality of the model. Two different ways of verification are as follows: (a) by data on dynamics of the study object excluded from identification, or (b) by data on another similar object. In our case of 34 simultaneously analyzed river basins, the latter is more appropriate. For instance, if any of the 34 river basins are excluded from identification, we get practically the same model parameters, as in the case of 34 basins. It is obvious that discrepancy of calculations also remains the same when verifying the model for the excluded basin.

Discrepancy between the calculated and observed values always exists due to errors in experimental measurements of characteristics of the modeled object, their changes in time and space, as well as limited adequacy of the model itself. An important feature of

SAM is the reduced sensitivity to initial data error. A large error only increases discrepancy not changing either the sought equations or their parameter values.

At the third stage, the degree of adequacy and sensitivity of the developed model to environmental factors is assessed. We propose the simple criterion for adequacy $A$ of mathematical models [24,30]:

$$A = \frac{1}{\sqrt{2}} \frac{S_{\text{dif}}}{S_{\text{obs}}}, \tag{2}$$

where $S_{\text{dif}}$ is the standard (root mean square) deviation for the difference between calculated and observed values of the output variable (model residual); $S_{\text{obs}}$ is the standard deviation for observed values of the output variable; and $1/\sqrt{2} = 0.71$ is a multiplier.

According to (2), $A$ is the model error normalized to the standard deviation of observed data. The interval 0–0.71 characterizes model adequacy varying between the best at $A = 0$ and acceptable at $A = 0.71$. The value of $A = 0.71$ corresponds to the simplest model consisting of one number, i.e., the average of the modeled characteristic. Criterion $A$ is similar and related to the *RMSE*-standard deviation ratio (*RSR*) [31,32] and the Nash–Sutcliffe model efficiency coefficient (*NSE*) [32] via dependences $RSR = A\sqrt{2}$ and $NSE = 1 - RSR^2 = 1 - 2A^2$.

The *FS* criterion calculated by using $A$ characterizes the model sensitivity to natural variations in environmental factors [30,33]:

$$FS = \left(A'\right)^2 - \left(A\right)^2 = \frac{\left(S'_{\text{dif}}\right)^2 - \left(S_{\text{dif}}\right)^2}{2(S_{\text{obs}})^2} = \frac{2(S_{\text{fac}})^2}{2(S_{\text{obs}})^2} = \frac{(S_{\text{fac}})^2}{(S_{\text{obs}})^2}, \tag{3}$$

where *FS* is the model sensitivity to input environmental factor; $A'$ is the value of $A$ obtained by (2) after substituting into the model randomly mixed-up observed values of factor (which obviously have the same statistical distribution and variance); $(S_{\text{dif}})^2$ is the variance for the difference between calculated and observed values of output variable (model residual variance); $(S'_{\text{dif}})^2$ is the same variance at substitution of mixed-up observed values of rated environmental factor into the model; $(S_{\text{fac}})^2$ is the contribution of natural variations in rated factor to the variance of output variable; and $(S_{\text{obs}})^2$ is the variance of observed values of output variable (in our case, spring flood).

In accordance with Equations (2) and (3) and the variance sum law, $(S'_{\text{dif}})^2$ exceeds $(S_{\text{dif}})^2$ by two equal variances $(S_{\text{fac}})^2$. The latter are (a) the variance from the contribution of actual variations in the input factor to observed output variable and (b) the variance from the contribution of the same but randomly mixed variations to calculated output variable. For adequate models, variances "a" and "b" are absent in $(S_{\text{dif}})^2$ because of subtraction of the calculated and observed contributions of the input factor. Since the variance due to observation errors of the input factor is present in $(S'_{\text{dif}})^2$ and $(S_{\text{dif}})^2$, its values are subtracted from each other in (3). Therefore, excluding the observation errors from (3), we assess the model sensitivity *FS* directly to natural variations in the input factor. If this factor is accounted incorrectly or absent in the model equations, *FS* is zero. When description of its influence on the characterized processes is accurate, the *FS* value reaches its maximum. Obviously, with a single-valued correspondence between factor values and model output variable, the sensitivity of *FS* will reach its theoretical maximum of 1.

In contrast to time-consuming global [34] or local [35,36] sensitivity analyses of mathematical models, computation of sensitivity *FS* is easily performed based on Equations (2) and (3). Estimation of *FS* requires only a nonrecurrent model calculation with randomly mixed values of a chosen input factor. In addition, *FS* characterizes the simulated process sensitivity directly to *natural variations of factors* since *FS* excludes related observation errors. For example, our assessing spring flood sensitivity to meteorological factors eliminates the errors of their spatial generalization. Similarly, we exclude objectively large errors of landscape boundaries and areas (determined from 1:200,000 scale maps) from flood sensitivity to landscape structure of river basins. Other well-known methods for assess-

ing sensitivity cannot obviate the influence of input data errors; they understate the real sensitivity of simulated processes to environmental factors. In case of mountainous areas, these methods routinely underestimate the real sensitivity of river runoffs to meteorological factors because of inaccuracy of reanalysis data.

*FS* allows us to estimate the relative importance of input factors and express it as a percentage of variance of output variable $(S_{\text{obs}})^2$ via multiplying *FS* in (3) by 100. As with *A*, *FS* can be easily expressed in terms of *RSR* or *NSE* [30]: $FS = [(RSR')^2 - (RSR)^2]/2$ and $FS = (NSE - NSE')/2$. Here, $RSR'$ and $NSE'$ are calculated similarly to $A'$ in (3).

## 2.3. Advantages of System-Analytical Modeling

Summarizing the aforesaid, we suggest considering the main advantages of the SAM methodology through comparison with other modeling methods of complex hydrological systems. First of all, SAM is the system approach that gives an adequate description of insufficiently studied processes of flood formation in the mountains as well as an estimation of the influence of related environmental factors. SAM is able to characterize processes if experimental data are unavailable. Although we had no data on water flows from landscapes, we put landscape runoff computations into the basis of the developed process-driven flood model, which describes real hydrological processes in watersheds and provides a further in-depth study of the analyzed hydrological systems (e.g., hydrochemical runoff from each landscape [24]).

Another advantage of SAM is the concurrent analysis of a large number of similar natural systems [19,20]; it ensures the universality of the found process dependences on environmental factors. To do this, the corresponding characteristic (monthly values of meteorological factors and river runoffs) is normalized to its long-term average in each analyzed system, and the normalized data are combined into a general sample for subsequent analysis [24,27,30]. To describe the given system, we should specify the values of only a few parameters of the model.

The SAM methodology involves specially developed methods for processing observation data on environmental factors:

- The identification of typological groups of geosystems (13 types of landscapes in Table 1) as autonomous hydrological subsystems of river basins with regard for their altitude belt and structural-tier heterogeneity, which is based on ArcGIS processing of cartographic materials of scale 1:200,000 [25];
- The normalization and spatial generalization of average monthly air temperatures and monthly precipitation for the mountainous areas according to the data from rare reference weather stations [26,27]. As a result, the orographic characteristics of river basins have practically no effect on the long-term dynamics of these factors and normalized river runoffs [33]. It should be noted that the reanalysis data for the Altai-Sayan mountain country are hardly suitable for building high-performance runoff models because of a sparse network of weather stations (11 reference weather stations involved in our study are located on the territory of 2,000,000 km$^2$) and change in precipitation up to an order of magnitude with variations in altitude [29];
- The application of free-form function *H* (Figure 4) able to adapt to real dependence of process on environmental factors. On the contrary, fixed forms of equations usually prevent models of complex natural systems from best performance.

SAM also includes the simplest method for assessing model sensitivity to natural variations of environmental factors and the model uncertainty quantification (evaluation of model residual variance components) [33]. Unlike all existing methods, we identify the simulated process to real changes in the factors, excluding observation errors. The model uncertainty quantification makes it possible to estimate the error of model equations themselves without factor-related errors, which is hardly feasible to implement via *RSR*, *NSE*, and other criteria.

In terms of applicability, SAM allows us to extract the particular information on functional relationships between processes and environmental factors from the long-term experimental data series. Unlike process- and data-driven models, SAM makes it possible to avoid costly field studies. For example, a complex of universal high-performance models for annual and long-term dynamics of hydrological and seven hydrochemical ($NO_2^-$, $NO_3^-$, $NH_4^+$, $PO_4^{3-}$, total dissolved iron, and dissolved and suspended matter) runoffs from each landscape of 34 river basins was developed using the observation data on river flows and substance concentrations in river water [24]. Consequently, complicated hydrological studies of individual landscapes in conditions of dissected mountain relief were excluded.

## 3. Results

According to SAM requirements, the number of observations of output variable dynamics should be ten times greater than that of model parameters. In our study, there were 1747 April runoffs for 34 rivers of the Altai-Sayan mountain country that theoretically allowed introduction up to 170 parameters into the developed model. The parameters of the models tested during SAM were identified via solving the inverse mathematical problem for the systems, including about 1000 yearly river runoff equations. For this purpose, calculated runoff was replaced by observed ones in the corresponding river basins.

After testing different equations describing the influence of meteorological conditions of the previous autumn and winter on April floods, we defined the balance model with least residual, which adequately takes into account the landscape structure of 34 river basins. It is similar to the previously developed predictive model of the entire spring–summer (for April–June) river runoff [30] and involves 39 parameters:

$$Q^i = H(c_1, c_2, 1, 1, c_3, c_4, P_1)\{\sum_k a_k S_k^i P_1 H(c_9, c_{10}, 1, 1, c_{11}, c_{12}, h_k^i) + \\ + \sum_k b_k S_k^i P_2 H(c_5, c_6, 1, 1, c_7, c_8, T_2) H(c_9, c_{10}, 1, 1, c_{11}, c_{12}, h_k^i)\} + d, \tag{4}$$

where $Q^i$—the predicted normalized runoff in April for the outlet of basin $i$, $i$ = 1–34; the first and second sums in the right part of (4) correspond to contributions of the recent autumn period (IX–XI months) and the current winter one (XII–III months), respectively; $a_k$, $b_k$—the parameters characterizing $k$-th landscape contributions to river runoff for the relevant period (Table 2), $k$ = 1–13; $S_k^i$—the relative area of $k$-th landscape of basin $i$; $h_k^i$—the landscape elevation, m a.s.l.; $P_1$, $P_2$—the mean deviations of normalized monthly precipitation from 1 (value 1 is the long-term average of normalized factor) in recent autumn and current winter periods, respectively; $T_2$—the mean deviation of normalized monthly air temperature from 1 in the current winter period; $H$—the piecewise–linear function (1); $c_{1-4}$, $c_{5-8}$, $c_{9-12}$—the parameters describing the influence of autumn precipitation $P_1$ and winter temperature $T_2$ on runoff volume in April as well as landscape elevation $h_k^i$ on precipitation amount; and $d$—the constant fraction of normalized runoff ($d \leq 1$) equal for all river basins, which depends on flow loss into soils and fractured rock zones.

The right side of (4) summarizes the contributions to runoff $Q^i$ in basin $i$ from each landscape. The contributions of landscapes in the first summand of Equation (4) depend on precipitation $P_1$ (soaked the soils going into winter) of the previous autumn, as well as $S_k^i$ and $h_k^i$ of landscapes. In the second summand in (4), landscape contributions are provided by winter precipitation $P_2$ and depend on $S_k^i$, $h_k^i$, autumn precipitation $P_1$, and winter temperature $T_2$, which affects moisture evaporation from the snow cover surface. Multiplier $H(c_1, c_2, 1, 1, c_3, c_4, P_1)$ in (4) accounts for moisture exchange between soils and snow cover in winter depending on soil moistening by autumn precipitation $P_1$ [37].

Model (4) was verified by excluding any of the 34 river basins from the identification stage with further testing of the model against observed runoff from this basin. From 1951 to 2020, we singled out different 33-year periods of identification, where the number of river runoff observations exceeded the number of parameters by an order of magnitude. This choice was due to the previously established patterns of climate change and possibility of performing various statistical estimates for these periods [38]. Testing implemented for different basins indicates that the discrepancy between calculated and observed flows was, on average, close to that obtained during identification that confirms the model universality.

**Table 2.** Contributions of landscapes to April monthly flood on rivers of the Altai-Sayan mountain country.

| Landscapes (Geosystem Groups) | Contributions ($a_i$, $b_i$ in Equation (4)) | |
| --- | --- | --- |
| | *a* | *b* |
| 1. Glacial–nival high mountains with permafrost | 0.06 | 0 |
| 2. Goletz alpine-type high and middle mountains, pseudogoletz low mountains with permafrost | 0.12 | 0.08 |
| 3. Tundra–steppe and cryophyte–steppe high mountains with permafrost | 0.07 | 0 |
| 4. Forest high middle and low mountains | 0.41 | 0.56 |
| 5. Exposure forest–steppe and steppe high and middle mountains | 0.39 | 0.55 |
| 6. Forest–steppe, steppe low mountains and foothills | 0.73 | 0.17 |
| 7. Intermountain depressions with different steppes and forest–steppes | 0 | 0.07 |
| 8. Steppe and forest–steppe piedmont | 0.79 | 0 |
| 9. Nondrainable and intrazonal landscapes with partial permafrost | 3.55 | 0 |
| 10. Mountain river valleys | 0.23 | ~0 |
| 11. Lowland river valleys | 1.65 | 1.05 |
| 12. Forest high and piedmont plains | 0.66 | 0 |
| 13. Aquatic landscapes | 0 | 0 |

The adequacy of the predictive model (4) was assessed according to equation (2) using residual $S_{dif}$ and standard deviation $S_{obs}$ for the entire sample of normalized runoffs from 34 river basins. The adequacy criterion $A$ made up 0.66, being a bit lower than the threshold value of 0.71. This means that we managed to express the general course (related to temperature, precipitation, and altitude) of hydrological processes in the river basins of the Altai-Sayan mountain country through (4). $A = 0.66$ is rather poor because of influencing individual features of hydrological processes in each river basin. Apparently, $A = 0.66$ is formal and impractical since it corresponds to performance of model (4) with unchangeable parameter values applied to any mountain river. At the same time, $A$ can be reduced by adapting Equation (4) to a certain basin and considering its specifics via refining parameter values (see below).

Look at the model sensitivity $FS$ to variations in environmental factors. The obtained values of $FS$ are given in Table 3. The highest sensitivity of spring flood falls on landscape structure. This is due to a significant difference in processes of snow accumulation and melting in different landscapes (forests, swamps, and steppes). Winter temperatures $T_2$ and landscape altitude $h_k^i$ have the least effect on flood. The situation with $T_2$ is explained by the fact that incoming solar radiation, not air temperature, is responsible for most heat consumption needed for snow melting. The minimal altitude influence confirms adequate description of dynamics of meteorological factors via their normalized values. The latter hardly depend on the terrain altitude [30], while the altitude dependence of temperatures (in °C) and precipitation (in mm) is great [29,39].

**Table 3.** Sensitivity $FS$ [1] of model (4) to natural variations in environmental factors.

| Characteristic | Value |
|---|---|
| Adequacy $A$ of model (4) according to Equation (2) | 0.66 |
| Standard deviation [2] $S_{obs}$ of actual river runoff $Q$ | 0.44 |
| Sensitivity $FS_L$ to variations in landscape structure of river basins [3] | 49 |
| Sensitivity $FS_P$ to joint variations in autumn and winter precipitation ($P_1$ and $P_2$) | 14 |
| Sensitivity $FS_{P1}$ to autumn precipitation $P_1$ | 9 |
| Sensitivity $FS_{P2}$ to winter precipitation $P_2$ | 3 |
| Sensitivity $FS_T$ to winter air temperature $T_2$ | 0.7 |
| Sensitivity $FS_h$ to landscape elevation $h_k^i$ | <0.1 |

Notes: [1] estimated by (3) and expressed in percent of variance $(S_{obs})^2$; [2] calculated as mean standard deviation of normalized observed runoff in 34 river basins. At the same time, it corresponds to mean standard deviation in fractions (or as percentage when multiplied by 100%) of non-normalized observed runoff; and [3] calculated via joint random mixing values of landscape hydrological characteristics ($a_k$ and $b_k$ in (4)) among 34 basins, $k = 1–12$.

## 4. Analysis of Results and Discussion

Let us examine the relationship between April monthly floods and meteorological factors for the rivers of the Altai-Sayan mountain country. Figure 5 demonstrates the dependence of predicted April runoff of the river Katun on air temperature and precipitation for previous autumn and winter periods according to the model (4). We deal with a simple linear relationship between spring runoff and precipitation in both seasons (Figure 5a). The established relationship considerably differs from the more complex effect of precipitation on the entire spring–summer flood in April–June [30]. It is easily explained by partial thawing of soils in April that limits meltwater infiltration into soils [37] and supports direct runoff to rivers. The less the amount of autumn precipitation (that soaks soils going into winter), the less the runoff is.

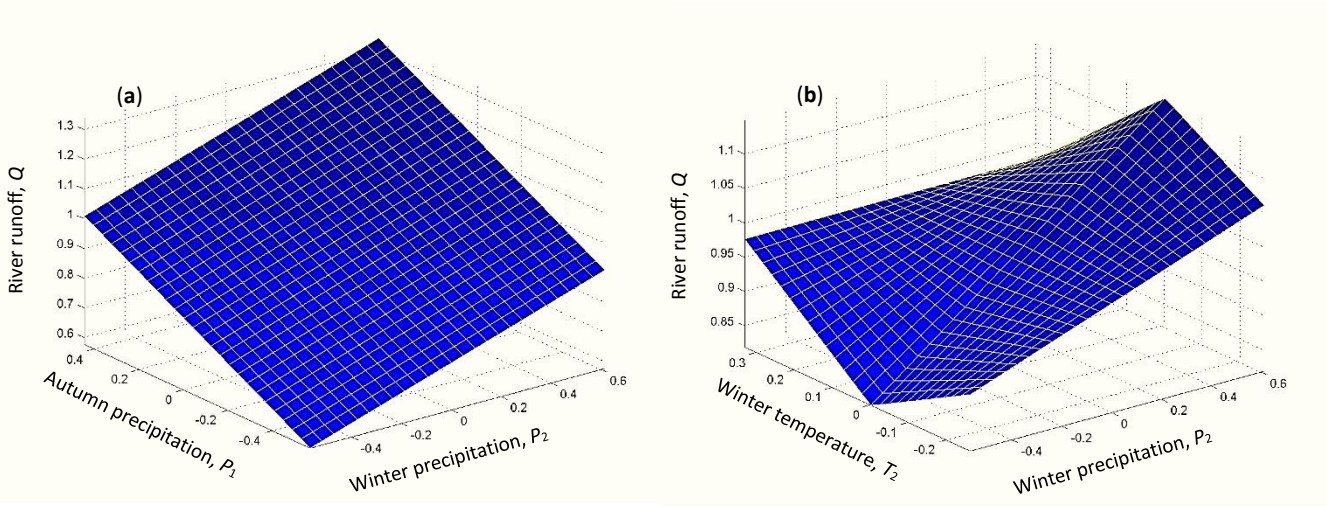

**Figure 5.** Dependence of April monthly floods (with ice motion) of the river Katun on air temperature and precipitation for preceding seasons (we show deviations of normalized monthly meteorological characteristics from their long-term averages, see notation for Equation (4)). (**a**) April runoff $Q$ as a function of precipitation $P_1$ (IX–XI) and $P_2$ (XII–III); (**b**) April runoff $Q$ as a function of temperature $T_2$ (XII–III) and precipitation $P_2$.

The dependence of river runoff on winter temperature and precipitation is given in Figure 5b. The influence of these factors is more intricate and similar to their effect on the entire spring–summer flood [30]. The dependence in Figure 5b suggests a more narrow range of flood changes that is again due to partial thawing of soils and a slightly varying area of snow cover in April. Solar radiation-induced melting of snow ensures more stable runoff of meltwater into rivers. At large amounts of winter precipitation $P_2 > 0$, fall in

winter air temperature $T_2 < 0$ results in lower temperature of thick snow cover, greater storage, and longer time lags for spring meltwater to travel through a snowpack [40] and move downslope into rivers. These processes bring to reduction in April runoff $Q$. With growing $T_2 > 0$, $Q$ drops again due to slower freezing of soils and more intensive moisture transition to lower soil layers and groundwater [41], resulting in soil insiccation in winter and greater loss of meltwater for soil soaking in spring.

At small winter precipitation $P_2 < 0$, April runoff $Q$ grows with air temperature drop $T_2 < 0$ in winter (Figure 5b). A thin snow cover contributes to the deep freezing of soils in winter (up to 2–3 m [37]). In spring, such a freeze facilitates ice layer formation in the upper soil layer and prevents meltwater infiltration into soil [37]. The lower the $T_2$, the larger the area of the ice layer is and the more meltwater come into rivers. Rise in temperature $T_2 > 0$ also results in increased runoff $Q$ caused by more intensive melting of snow cover on slightly frozen soils at all altitudes. Such an increase in runoff $Q$ is limited by incoming solar radiation responsible for 50–80% of snowmelt and runoff formation [42].

Let us consider the possibility of practical application of the predictive model (4). Most often, a spring flood forecast is required for a certain river with the known data on its runoff, air temperature, and precipitation. In this case, it is possible to refine the model parameters and improve the accuracy of the resulting forecast.

Choosing a certain river, one can exclude the influence of unaccounted variations in landscape structure of river basins (blurring of altitudes, areas, and hydrological characteristics of landscapes) from the model (4). According to Figure 5a and Equation (4), the negligible dependence of runoff on autumn precipitation $H(c_1, c_2, 1, 1, c_3, c_4, P_1)$ also may be excluded. Then the predictive model of the April monthly flood takes the form:

$$Q = aP_1 + bP_2 H(c_5, c_6, 1, 1, c_7, c_8, T_2) + d, \tag{5}$$

where former designations of variables and parameters are used. In (5), parameters $c_{5-8}$ are already known from (4), while $a$, $b$, and $d$ must be recalculated from observed runoff, average monthly temperature, and monthly precipitation in the study basin. To do this, one can use a regression analysis or SAM.

Denote the adequacy criterion for Equation (5) as $A_0$. To find it, we evaluate the components of the residual variance $(S_{\text{dif}})^2$ for the model (4) through the method of model uncertainty quantification [33]. In accordance with the method, $(S_{\text{dif}})^2$ adds up the variance components formed by equation inaccuracy, observation errors of input factors and output variable, and unaccounted variations of input factors (blurring of landscape structure in our case) [30,33]. In $(S_{\text{dif}})^2$, we can select the components, which are absent in residual variance of the model (5), and estimate the remaining component that characterizes $A_0$. Given (2) and (3), we get:

$$2A^2 = (S_{\text{dif}})^2/(S_{\text{obs}})^2 \approx FS_L + FS_P \times 2A_P^2 + FS_T \times 2A_T^2 + 2A_0^2, \tag{6}$$

where $A$ is the adequacy of Equation (4) (Table 3), $(S_{\text{dif}})^2$ is the residual variance of Equation (4) (i.e., discrepancy between predicted by (4) and observed runoff), $(S_{\text{obs}})^2$ is the variance of observed runoff, $FS_L$ is the contribution from unaccounted variations in landscape hydrological characteristics $a_k$ and $b_k$ (Tables 2 and 3), $FS_P$, $FS_T$ are the contributions from precipitation variations and air temperature variations (Table 3), $2A_P^2$, $2A_T^2$ are the shares in variances of precipitation and air temperature variations, which are formed by errors of their spatial averaging, $A_P$, $A_T$ are the adequacy (2) for spatial averaging of the same meteorological factors, and $2A_0^2$ is the sought component of $(S_{\text{dif}})^2$ characterizing adequacy $A_0$ of Equation (5).

Substituting into (6) the values in fractions of unit $A = 0.66$, $FS_L = 0.49$, $FS_P = 0.14$, $FS_T = 0.007$ (Table 3), and averages $A_P = 0.73$ and $A_T = 0.32$ for autumn (IX–XI) and/or

winter (XII–III) months [26], we obtain $A_0$ for the model (5) describing an individual river basin:

$$2 \times (0.66)^2 \approx 0.49 + 0.14 \times 2(0.73)^2 + 0.007 \times 2(0.32)^2 + 2A_0^2 \quad \text{or} \quad A_0 \approx 0.34.$$

Overall, it is considered that $A_0 \approx 0.34$ characterizes adequacy of April monthly flood forecasts for any river basin of the Altai-Sayan mountain country. Indeed, a sample of normalized observed runoff from any typical basin and a similar sample of runoff from 34 basins will have the same (differing only in volumes) statistical characteristics. Therefore, $A_0$ values of forecast adequacy in both cases will be the same.

Let us estimate how variance $(S_{\text{dif}})^2$ for discrepancy of flood runoff predicted by (5) decreases as compared to variance $(S_{\text{obs}})^2$ corresponding to discrepancy of a trivial forecast by the long-term average observed runoff. From Equation (2) and $A_0 \approx 0.34$, we get a fourfold reduction in variance:

$$A_0^2 = (S_{\text{dif}})^2 / 2(S_{\text{obs}})^2 \approx (0.34)^2 \quad \text{or} \quad (S_{\text{dif}})^2 \approx 0.23(S_{\text{obs}})^2.$$

We can determine the quality of the model (5) by the Nash–Sutcliff coefficient $NSE_0 = 1 - 2A_0^2$ (see notation for Equation (2)). At $A_0 \approx 0.34$, $NSE_0 \approx 1 - 2(0.34)^2 \approx 0.77$. Note that just one mathematical criterion ($A$, $NSE$, $RSR$. $R^2$, etc.) is sufficient for assessing our process-driven model adequacy [33]. Process-driven models describe real physical–hydrological processes, and consistency between the obtained simulation results and the related scientific concepts serves as an extra confirmation of the model performance. In turn, data-driven models based on statistical techniques for data processing employ two or more different criteria [7,9].

The quality of April monthly flood forecasts with $NSE_0 \approx 0.77$ is in the best range ($0.75 < NSE \leq 1.0$) for mathematical models [32]. It is important that floods are predicted with such a "very good" quality for the mountain areas, the hydrological processes of which are extremely hard to simulate. Moreover, the obtained quality can be improved through application of the established dependence of spring floods on April meteorological conditions [20]. Such refinement of forecasts obtained from (5) is easily performed by substituting the available meteorological prognoses of air temperature and precipitation for April into this dependence.

Finally, it should be noted that models (4) and (5) are also intended to be included in the water quality hydrochemical model developed using SAM [24] in order to predict the content of $NO_2^-$, $NO_3^-$, $NH_4^+$, $PO_4^{3-}$, total dissolved iron, and dissolved and suspended matter in river water during spring floods.

## 5. Conclusions

1. Description of meteorological factors and river runoffs as a fraction of the corresponding long-term averages made it possible to unify their dynamics throughout the Altai-Sayan mountain country. The performed normalization and spatial generalization of average monthly air temperature and monthly precipitation over the entire territory of the country adequately reflect their long-term dynamics and ensure more accurate calculations of hydrological processes in contrast to in situ observations at rare weather stations;

2. The developed universal predictive model (4) adequately describes the influence of environmental factors on April monthly runoffs with ice motion for 34 rivers of the Altai-Sayan mountain country. It takes into account the landscape structure (geosystems) of river basins, monthly values of temperature and precipitation for the previous autumn and winter, as well as watershed altitudes. The obtained universal dependences of runoff on meteorological factors correlate with physical–hydrological patterns of snow cover melting, freezing and thawing of moistened soil (depending on snow cover thickness), ice layer formation in the upper soil layer (preventing meltwater infiltration into soil), and solar radiation effect. The reverse transition from

normalized river runoffs in (4), (5) to their measurement in m$^3$/s is easily performed through multiplication by the corresponding long-term average runoff of the basin. Our findings expand the existing notion about hydrological processes and factors, which determine the intensity of annual spring floods in the mountains;

3. The sensitivity of the developed model (April monthly runoff with ice motion) to natural variations in temperature and precipitation, landscape structure of river basins, and landscape altitude was estimated. In contrast to other methods for assessing model sensitivity, our methodology excludes the influence of observation errors of environmental factors. Using the method of model uncertainty quantification aimed at the estimation of all residual variance components, we evaluated the performance of a simplified version of the developed model applicable to any river basin;

4. The simplified model (5) provides a medium-term April flood forecast for any river basin in the Altai-Sayan mountain country or other mountainous areas. The forecast quality is characterized by the Nash–Sutcliff coefficient $NSE_0 \approx 0.77$ that is "very good" for hydrological models of mountain rivers. The value of $NSE_0$ can be further improved by means of additional consideration of standard meteorological prognoses of air temperature and precipitation for April. In the country under study, April runoffs turn occasionally into catastrophic floods [23]. Timeliness and high accuracy of forecasts are extremely important for decision making in ensuring local population safety and for the region administration provided with flood forecasts performed according to the developed methodology;

5. The proposed SAM methodology of building the process-driven hydrological (and hydrochemical) models provides an in-depth study of complex hydrological systems with a lack of information on their structure and functional relationships with environmental factors. Thus, this methodology can serve as an effective scientific tool for studying natural hydrological systems influenced by environmental and anthropogenic factors.

**Author Contributions:** Conceptualization, Y.B.K.; methodology, Y.B.K.; software, Y.B.K.; validation, Y.B.K.; formal analysis, Y.B.K. and I.A.T.; investigation, Y.B.K.; resources, Y.B.K. and I.A.T.; data curation, Y.B.K. and I.A.T.; writing—original draft preparation, Y.B.K.; writing—review and editing, Y.B.K.; visualization, Y.B.K.; supervision, Y.B.K.; project administration, Y.B.K. and I.A.T.; funding acquisition, Y.B.K. and I.A.T. All authors have read and agreed to the published version of the manuscript.

**Funding:** This research was financially supported by the Russian Science Foundation, grant number 22-27-00058.

**Data Availability Statement:** Some meteorological and runoff data used in this study are available at sites: All-Russian Research Institute of hydrometeorological information. World data center of GU VNIIGMI-MTsD—http://meteo.ru/ (accessed on 1 February 2023). Weather for 243 countries of the world (rp5.ru)—https://rp5.ru/Weather_in_the_world (accessed on 1 February 2023). Weather and Climate (Pogoda I Klimat)—http://www.pogodaiklimat.ru/archive.php (accessed on 1 February 2023). A Regional, Electronic, Hydrographic Data Network For the Arctic Region—https://www.r-arcticnet.sr.unh.edu/v4.0/index.html (accessed on 1 February 2023). Automated information system of state monitoring of water objects—https://gmvo.skniivh.ru/ (accessed on 1 February 2023).

**Conflicts of Interest:** The authors declare no conflict of interest.

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
