# Peer review of "High-Performance Forecasting of Spring Flood in Mountain River Basins with Complex Landscape Structure"

_water, doi:10.3390/w15061080_

Round 1

Reviewer 1 Report (New Reviewer)

This paper presents a methodology for building a process-driven model for medium-term prediction of spring floods in mountainous areas, where hydrological analysis is much more complex compared to plains. The approach is based on a systematic analytical model of complex hydrological processes in 34 watersheds of the Altai-Sayan mountain region. Considering 13 types of landscapes as autonomous hydrological subsystems affecting river runoff (1951-2020), the paper developed a generalized predictive model of the most dangerous April monthly runoff (ice movement) applicable to any watershed. The input factors to the model are monthly mean temperature and monthly precipitation for the most recent fall and current winter, as well as watershed landscape structure and topography data calculated through GIS tools. The description of meteorological factors and river runoff as part of the corresponding long-term averages makes it possible to unify their dynamics across the country.

The questions are as follows:

1. It is recommended that citation of references is not clustered together.

2. The indentation format in the conclusion is different from the previous text.

3. It would be more intuitive to divide the explanation of each parameter in the formula into one paragraph.

4. It is suggested to add a paragraph in the conclusion to summarize the whole text.

Author Response

Response to the first reviewer

Dear reviewer,

Our revisions are as follows (the edited text is highlighted in yellow):

  1. It is recommended that citation of references is not clustered together” – Cluster [1-5] has been split (first paragraph of the text).
  2. The indentation format in the conclusion is different from the previous text” – Revised in line with the previous text.
  3. It would be more intuitive to divide the explanation of each parameter in the formula into one paragraph” – Revised (comments to Eq. 2).
  4. It is suggested to add a paragraph in the conclusion to summarize the whole text” – One paragraph has been added in the conclusion.

Thank you very much for your fair comments.

You wrote:

The questions are as follows:

  1. It is recommended that citation of references is not clustered together.
  2. The indentation format in the conclusion is different from the previous text.
  3. It would be more intuitive to divide the explanation of each parameter in the formula into one paragraph.
  4. It is suggested to add a paragraph in the conclusion to summarize the whole text.

Reviewer 2 Report (New Reviewer)

This is an interesting paper with novelty/significance for publication. I recommend accepting with minor comments (see attached file).

Author Response

Response to the second reviewer

Dear reviewer,

Our revisions are as follows (the edited text is highlighted in yellow):

  1. “…the hydrological analysis of which is much more complicated in contrast to plains.” (reviewer: This may no always be true) – Revised (see the abstract).
  2. “The input factors of the model are the average monthly air temperature and monthly precipitation for recent autumn and current winter periods” (reviewer: It is not clear which month/season are they talking. Are all preceding seasons?) – Revised (see the abstract).
  3. Plot the map in relation to the country and continent” – Figure 1 has been revised.

Thank you very much for your review.

Reviewer 3 Report (New Reviewer)

The manuscript deals with a very interesting topic. However, some revisions are needed before publication.

The abstract is not clear enough and is too long, it needs to be rewritten and summarized.

The coefficients a and b shown in Tab. 1 should be moved to a new table to be added after the definition of eq. 4.

Paragraph 2.2 is too long and should be split into two or more subparagraphs.

The methodology is complex and a flow chart should be added to improve readability.

Paragraph 3 "Results" contains too much text, graphs and tables should be added to improve readability.

The results shown in section 4 should be moved to section 3, focusing section 4 on discussion only. Furthermore, the discussion should be enhanced by highlighting the limitations of the model and explaining how it could be applied in other contexts.

Author Response

Response to the third reviewer

Dear reviewer,

In response to Tanner Li “Please revise the manuscript according to the referees' comments and upload the revised file within 7 days”, we would like to inform you that being pressed for time we have done our best to edit the text in line with all comments and recommendations of the three reviewers. Thank you for understanding.

Our revisions (the edited text is highlighted in yellow):

  1. The abstract is not clear enough and is too long, it needs to be rewritten and summarized.” – We have analyzed the abstract anew and removed 3 sentences.
  2. The coefficients a and b shown in Tab. 1 should be moved to a new table to be added after the definition of eq. 4.” – Table 2 has been added.
  3. Paragraph 2.2 is too long and should be split into two or more subparagraphs.” – Done (paragraph 2.3.)
  4. The methodology is complex and a flow chart should be added to improve readability.” – Created (Fig. 3).
  5. Paragraph 3 "Results" contains too much text, graphs and tables should be added to improve readability.” – Table 2 has been added.
  6. The results shown in section 4 should be moved to section 3, focusing section 4 on discussion only. Furthermore, the discussion should be enhanced by highlighting the limitations of the model and explaining how it could be applied in other contexts.” –

In this section we do not consider the equations developed (i.e., the model itself), which are the main result of our research. We just explain the obtained influence of factors on floods through its comparison with the experimental data available from the literature.

Our model was developed by the example of the Altai-Sayan mountainous country with a very contrasting climate, ranging from a cold deserts at high altitudes to forests and steppes at lower ones. In our opinion, there are no significant limitations in applying the proposed methodology and model in practice. Note that the SAM methodology has allowed us to develop the process-driven models of high performance for:

(a) moisture exchange in soils of Russia and the United States [https://doi.org/10.1016/j.ecolmodel.2005.05.028],

(b) phenological development of wheat plants [https://doi.org/10.1016/0304-3800(94)90137-6],

(c) grain yields in agroecosystems of Russia and the United States [https://doi.org/10.1016/j.ecolmodel.2005.05.027],

(d) content of hazardous pollutants in grain [https://doi.org/10.18799/24131830/2022/9/3699 , https://doi.org/10.25743/SDM.2021.58.67.055 ],

(e) climate change in Russia and the United States [https://doi.org/10.1016/j.ecolmodel.2005.05.027 , https://www.scopus.com/sourceid/21100218356 ],

(f) emergence of climate destabilization zones on the continents [https://applied-research.ru/ru/article/view?id=5278],

(g) hydrological flows of dozens of mountain rivers [https://doi.org/10.32523/2306-6172-2020-8-2-69-85],

(h) seven hydrochemical ( , , total dissolved iron, dissolved and suspended matter) runoffs from the same rivers [https://doi.org/10.1007/978-3-030-57488-8_7],

(i) evaluation of the quality  of mathematical models [https://doi.org/10.32523/2306-6172-2020-8-3-67-77]

We also explain how our predictive model could be applied in other contexts (for individual basins) and show its accuracy. Following your recommendation, we added one paragraph “Finally, it should be noted that models (4) and (5) are also intended to be included in the water quality hydrochemical model developed using SAM [24] in order to predict the content of , total dissolved iron, dissolved and suspended matter in river water during spring floods” at the end of section 4.

Thank you very much for your careful review.

You wrote:

The manuscript deals with a very interesting topic. However, some revisions are needed before publication.

The abstract is not clear enough and is too long, it needs to be rewritten and summarized.

The coefficients a and b shown in Tab. 1 should be moved to a new table to be added after the definition of eq. 4.

Paragraph 2.2 is too long and should be split into two or more subparagraphs.

The methodology is complex and a flow chart should be added to improve readability.

Paragraph 3 "Results" contains too much text, graphs and tables should be added to improve readability.

The results shown in section 4 should be moved to section 3, focusing section 4 on discussion only. Furthermore, the discussion should be enhanced by highlighting the limitations of the model and explaining how it could be applied in other contexts.

Round 2

Reviewer 3 Report (New Reviewer)

The authors have significantly revised the manuscript and satisfactorily responded to the reviewers' comments. Therefore, it is possible to accept the manuscript in present form.

This manuscript is a resubmission of an earlier submission. The following is a list of the peer review reports and author responses from that submission.

Round 1

Reviewer 1 Report

Revision of “High-performance forecasting of spring flood in mountain river basins with complex landscape structure”

The authors present a “model for a medium-term forecast of spring floods based on autumn-winter temperatures and precipitation” for which they proposed a conceptual scheme that is a simplified version of a surface runoff (“an adequate reflection of hydrological processes in the mountain geosystems”). This equation 4 includes an important number of parameters that were fitted to the observed data.

The overall impression of the manuscript is that there is a lot of missing information that makes difficult to evaluate and to replicate this model in other conditions. For example, and I apologise for such a basic question: when the authors say “spring floods”, are they referring to the peak flow during floods, the mean flow of April, or the daily average flow during a flood event?. The same information is missing for precipitation and the rest of input variables.

Furthermore, a standard graphic comparison between observed and predicted flow is needed. In particular, if I’m correct, the value A=0.66 of the adequacy of model defined in Eq. 2 is associated to Nash–Sutcliffe model efficiency coefficient equal to 0.135, which usually is not an acceptable value.

Reviewer 2 Report

The manuscript “High-performance forecasting of spring flood in mountain river basins with complex landscape structure” proposes a model for flood forecasting, using a so-called “system-analytical modeling” approach. Unfortunately, despite the importance of flood forecasting research, the originality of the proposed method and presentation of the research are not meeting the journal’s standards for publication. Please see my details below:

1.     Originality: The biggest flaw of the paper is the method itself is too trivial and traditional to be considered as original research. The most important equation for the proposed method is equation (1), in which multiple parameters and input factor’s dependencies are characterized with linear combinations. This is one of the most traditional ways of doing statistical modeling to model the dependency among variables. With recent advances in statistical learning/machine learning/deep learning, there are far more advanced methods with much better performance. Besides, the authors should also cite the most recent research in flood forecasting. The literature review is very limited.

2.     Self-citation: Another big issue of the paper is that self-citation is too high. Line 117 (Section 2.2) cited four papers about the “System-Analytical Modeling” (SAM) method, which were all the authors’ own publications. There are two things that the authors should address when citing their own research: a. What’s new (or different) in this paper? b. Does any other researcher support or has cited the proposed SAM method?

3.     Experimental design: The authors didn’t compare the proposed method with any of the state-of-the-art methods, but still claims the proposed model is unique (line 22) and universal with high accuracy (line 61, line 369, line 372). The method is not “unique” and there is no “universal” method in the world. The accuracy is only meaningful when compared with other methods. 

4.     Presentation: Too few results were shown in the study to prove the effectiveness or accuracy of the method. Tables and figures contain very little information, and none of them can show how good the proposed method is. The conclusions (Section 5) cannot be supported by the experiment at all.